# Signaling Pathways Controlling Axonal Wrapping in *Drosophila*

**DOI:** 10.3390/cells12212553

**Published:** 2023-10-31

**Authors:** Marie Baldenius, Steffen Kautzmann, Suchet Nanda, Christian Klämbt

**Affiliations:** Institute for Neuro- and Behavioral Biology, Faculty of Biology, University of Münster, Röntgenstraße 16, D-48149 Münster, Germany; m_bald03@uni-muenster.de (M.B.);

**Keywords:** *Drosophila*, wrapping glia, axon, myelin, receptor tyrosine kinase signaling, cell–cell adhesion

## Abstract

The rapid transmission of action potentials is an important ability that enables efficient communication within the nervous system. Glial cells influence conduction velocity along axons by regulating the radial axonal diameter, providing electrical insulation as well as affecting the distribution of voltage-gated ion channels. Differentiation of these wrapping glial cells requires a complex set of neuron–glia interactions involving three basic mechanistic features. The glia must recognize the axon, grow around it, and eventually arrest its growth to form single or multiple axon wraps. This likely depends on the integration of numerous evolutionary conserved signaling and adhesion systems. Here, we summarize the mechanisms and underlying signaling pathways that control glial wrapping in *Drosophila* and compare those to the mechanisms that control glial differentiation in mammals. This analysis shows that *Drosophila* is a beneficial model to study the development of even complex structures like myelin.

## 1. Introduction

The development and function of the nervous system relies on a proper interplay between its major cell types, neurons, and glial cells. Neurons form large networks to compute information and communicate with their targets. Information is generally perceived at apical dendrites and integrated towards the axon hillock where it is transformed to trains of action potentials that travel down the axon. Glial cells are needed at many levels in the brain and are essential for its formation and function. They regulate neuronal cell number, metabolically support neuronal activity, and control synaptic transmission and plasticity. By regulating the radial diameter of the axon and the distribution of voltage-gated ion channels along this cell process, glial cells also influence conduction velocity. Not surprisingly, glial morphology reflects these functional specializations.

The differentiation of glial cells that can wrap axons requires a complex set of neuron–glia interactions to ensure that glial cells identify axons and are able to select the ones they will eventually wrap. They need to form an insulating coat around groups or single axons, and thus invest heavily in the synthesis of membrane lipids [1,2]. The ability of glia to recognize the axon, grow around it, and eventually arrest this growth likely depends on the integration of several signaling systems. In humans, any disruption of these processes, such as those seen in Charcot Marie Tooth disease, has detrimental effects on health. Therefore, a thorough understanding of the interactions between neurons and glia is essential and an analysis of this interaction in simpler organisms might help to identify new players and components regulating the correct differentiation of wrapping glia.

Here, we put *Drosophila* on the center stage and summarize what is known about the pathways controlling glial wrapping. We consider wrapping as the planar growth of glial processes around axons or axon fascicles, which eventually results in insulation and the generation of specific compartments. We discuss how wrapping of axons occurs in flies and how wrapping is controlled at the molecular level. The comparison with what is known about mammalian glial differentiation suggests that flies indeed can pose as a beneficial model to study the development of even complex structures like myelin.

## 2. *Drosophila* as a Model to Study Glia Development

The *Drosophila* nervous system, though simpler in organization compared to mice or humans, harbors different types of neurons and glial cells. Most of the functional properties that were attributed to the glia of the vertebrate nervous system are also associated with the *Drosophila* glia. However, compared to vertebrates, the total number of glial cells in *Drosophila* is very low which facilitates the detailed analysis of neuron–glia interactions [3]. Moreover, work in the last two decades has demonstrated that not only functional properties of glia, but also the underlying molecular signaling cascades, are often conserved between the animal kingdoms [4]. *Drosophila* is a holometabolous insect, and larval and adult stages are both characterized by different needs regarding their nervous systems. 

Embryonic development generates a relatively slow-moving larva, with a simple central nervous system (CNS) harboring very few glial cells [5,6,7,8]. The adult fly brain, which is generated during pupal stages, is much more complex to match the output requirements needed to control flight or complex leg movements [9,10]. Correspondingly, the number of glial cells in the CNS and the peripheral nervous system (PNS) is much larger compared to larval stages [11,12]. In addition, the functional properties of glia might differ between larval and adult stages to provide the basis for more complex computing and faster transmission velocities. 

## 3. The Different Glial Cell Types Wrapping Axons in the *Drosophila* Nervous System

The larval nervous system is formed during embryonic development by a limited number of stem cells that generate a highly stereotyped pattern of neurons and glial cells. The lineages of the larval glia have been established [5,7]. In each abdominal neuromere, 65 glial cells are found next to 700 neurons. Seven principal glial cell types are known in the larval nervous system [3]. Three of these glial cell subtypes display the ability to wrap around axons (Figure 1 and Figure 2).

The midline glial cells wrap axons running in the two segmental commissures and the large brain commissure in the larval CNS [14,15,16,17]. These glial cells are special as they originate from the mesectoderm and lack expression of the transcription factor Repo, which is found in all other glial cell types. During the pupal stage, they undergo apoptosis, and no similar glial cell type is found in the adult CNS [15,18].

Ensheathing/wrapping glial cells encapsulate the neuropil and wrap around axons that enter or leave the neuropil in the larva [19,20,21] (Figure 1). The glial cells that cover the axons connecting peripheral and central nervous system in the adult fly are called tract glial cells [12] (Figure 2). Together, these glial cells might be considered the oligodendrocyte equivalent of the fly.

Finally, the wrapping glial cells wrap axons in the periphery [6,22,23]. In larvae, three wrapping glial cells cover each of the up to 4 mm long segmental nerves (A1-A7) and four wrapping glial cells cover the A8 nerve [6,24]. Initially, the wrapping glia extends long and thin processes along axons which eventually grow around single axons or groups of axons [6]. They represent the fly Schwann cell analog, mostly resembling non-myelinating Schwann cells found in Remak fibers [22,25]. In adults, many more wrapping glial cells are found [26]; however, the exact number has not yet been determined. Interestingly, these wrapping glial cells can form myelin-like structures [26].

In addition to these types of axon wrapping glial cells, four additional glial classes have been identified in *Drosophila*: The perineurial and subperineurial glia that constitute the blood–brain barrier, the cortex glia cells that engulf all neuronal cell bodies, and the astrocyte-like glial cells that send processes into the neuropil to regulate synapse function [3,27,28,29,30].

## 4. Wrapping Glia in the Peripheral Nervous System

The highly stereotyped larval segmental nerves undergo major remodeling in pupal stages. The larval motor neurons retract their axons, while the five posterior-most nerve pairs fuse to a single nerve bundle called the terminal nerve trunk [31]. Concomitantly, the larval wrapping glia dedifferentiates [31]. During midpupal stages, the number of wrapping glial cells increases in fused as well as unfused nerves, but the involved progenitor cells are not known [31]. Possibly, they originate from perineurial glial cells at the outer surface of the nerve, which would require these cells to migrate through an intact blood–brain barrier, or the dedifferentiated wrapping glia might divide, which appears unlikely as these cells are highly polyploid, similar to larval ensheathing and subperineurial glia [21,32].

In contrast to the abdominal nerves, other adult nerves are newly formed during pupal stages, such as the leg, optic, and wing nerves [13,33]. The three leg nerves (prothoracic, mesothoracic, and metathoracic leg nerve) harbor sensory and motor axons. In contrast, the optic nerve that is established at the end of larval life, harbors only sensory axons. The motor neurons that innervate muscles moving the *Drosophila* compound eye have different trajectories [34,35]. The wing nerve also harbors only sensory axons, but it branches off from the dorsal mesothoracic nerve that also harbors motor axons [36]. The leg and optic nerves are populated by glial cells originating from the CNS, whereas the wrapping glial cells of the wing nerve originate in the periphery [11,37,38,39].

During the formation of leg nerves in pupal stages, glial progenitors—that are not yet identified—generate about 300 glial cells that populate the 3 mm long leg nerve. In addition to these newly generated glial cells, one large polyploid wrapping glial cell is found on the later leg nerve, which likely is generated already during embryonic stages [11]. A typical adult leg nerve harbors around 760 axons [26]. The wrapping glial cells engulf these axons according to their size. While small caliber axons are generally wrapped as fascicles, larger axons are wrapped individually [26] (Figure 2).

The optic nerve is distinct from the leg nerve as it only harbors axons of sensory neurons. The compound eye comprises about 800 single eye units called ommatidia [40]. Within each ommatidium, eight photoreceptor neurons project their axons in a common fascicle towards the visual centers in the brain. The eight photoreceptor axons are engulfed as a fascicle by wrapping glial cells, which originate from perineurial glia of the CNS [39].

Similarly, the wing nerves harbor only axons of sensory neurons. In each nerve, 290 axons are engulfed as large fascicles. Only one central very large axon is engulfed by a single glial sheet [41]. The wrapping glial cells of the wing nerve originate from sensory organ precursors in the periphery and move in chains towards the CNS [37,38,42,43].

## 5. Wrapping Axons in the Central Nervous System

Although most axons in the peripheral nervous system are wrapped by glial processes, wrapping is scarce in the CNS. Serial electron microscopic imaging did not reveal any clear wrapping structures within the neuropil of the ventral nerve cord or the brain lobes [8,9,10]. The only three notable cases of CNS axon wrapping are found in areas lacking dendrites and synapses: Commissural axons link the two hemispheres and are wrapped by midline glia in the larval CNS [15,16,17]. A similar glial cell type has not yet been described in the adult nervous system. The axons that connect the brain lobes with the ventral nerve cord in the so-called neck region of the adult CNS are wrapped by tract glial processes [12]. These cells are not present in larvae. Finally, axons that connect the periphery with the central neuropil and vice versa are wrapped by ensheathing glial cell processes in the larval or by tract glial cell processes in the adult nervous system [12,26].

Glial wrapping in the CNS/PNS transition zone of the adult nervous system is different compared to wrapping of axons in the CNS/PNS transition zone of the larval nervous system [26]. At the CNS/PNS boundary of adult flies, large caliber motor axons show a pronounced axon initial segment of about 50 µm in length that is characterized by a high density of voltage-gated sodium and potassium channels [26]. Interestingly, the CNS-born tract glial cells precisely cover this axon initial segment that had been shown for single motor neurons before [44,45]. They originate from a position that in the larval CNS harbors the ensheathing/wrapping glial cells [12,19]; however, the lineage relationship between the two glial cell types is not yet established. The tract glia forms a special sponge-like meshwork along the axon initial segment. This generates a large lacunar system, possibly providing an interstitial fluid volume separating individual large caliber axons [26]. Distal to the lacunar system, the glial membrane sheets surrounding the axon initial segments appear to collapse, resulting in the formation of myelin-like sheets [26].

## 6. Different Phases during Wrapping Glia Differentiation

In order to disentangle the molecular processes that drive the differentiation of wrapping glia, it is helpful to reduce the complexity of the process and divide it into discrete steps. We propose that the differentiation of all wrapping glial cells can be divided into distinct phases that most likely occur separately in time. First, glial cells need to recognize their target axons (Figure 3A). Second, glial processes must grow along and around the axon or the axon fascicle to completely engulf the axon (Figure 3B). Third, growth around the axon needs to be terminated to ensure the single glial wrap (Figure 3C).

We propose that these basic processes are explained by differential axon–glia interactions that affect the regulating growth of signaling cascades. Glial growth towards an axon could be mediated by attractive, contact-independent signaling mechanisms such as fibroblast growth factor (FGF) signaling. Alternatively, explorative growth could be independent of an axon but upon neuron–glia contact, wrapping is initiated. We propose that the relative strength of this contact is decisive for the final mode of glial differentiation. 

Provided that axon–axon adhesion is stronger than axon–glia adhesion, groups of axons or fascicles are engulfed by a glial process as a unit (Figure 4A). This mode of wrapping is found in the developing compound eye, where the axons of all eight photoreceptor neurons housed in a single ommatidium are wrapped as a unit [46] or in the adult leg nerve where large groups of small diameter axons are engulfed by a single glial lining [26].

When axon–axon adhesion is weaker than axon–glia adhesion, single axons can be wrapped by single glial cell processes (Figure 4B). This type of wrapping predominates in the larval abdominal nerves, but is also found in the adult leg nerves where large caliber axons usually show a single glial wrap [6,26]. 

Finally, when glia–glia adhesion is equal to or stronger than axon–glia adhesion, wrapping is not stopped and multilayered glial membrane sheets form around an axon (Figure 4C). This wrapping mode is observed only in adult flies, where single large axons or groups of medium-sized axons are covered by multiple membrane sheets that are either embedded in a large extracellular volume to form so-called lacunar structures or are collapsed to form myelin-like structures [26]. Finally, it should be noted that glial growth could also occur in a purely cell-autonomous way. In the absence of axon–glia adhesion, growth is independent of axon–glia interaction and would result in stacked glial cell layers.

## 7. FGF Receptor Activation Controls Wrapping Glia Development

Work in the last decades has provided some understanding on how all the differentiation aspects are molecularly controlled. First, studies addressing the molecular control of wrapping glial differentiation focused on the developing visual system of the fly. The notion that the onset of glial differentiation coincides with photoreceptor axon–glia contact [39] promoted the search for receptor ligand pairs with a corresponding expression pattern. 

The FGF receptor, a member of the receptor tyrosine kinase (RTK) family, is expressed in many glial cells and is encoded by *heartless* (*htl*) [46,47,48,49]. The *htl* signaling is required in ensheathing glial cells, astrocytes, and wrapping glial cells of different nerves [20,23,46,48,50]. In all cases, *htl* function is linked to the extension of cell protrusions, and might represent the initial signal that triggers differentiation of wrapping glia (Figure 3A). In addition, it is still unknown whether signaling or adhesion systems at the axon-glia contact site may further define the specificity of wrapping. 

In the developing visual system, early activation of Htl is brought about by the FGF-8-like ligand Pyramus that is expressed by subperineurial glial cells. This renders the wrapping glia progenitors as mitogenic and motile [46,51]. The contact of wrapping glia progenitors with nascent photoreceptor axons that express the FGF-8-like ligand triggers a strong activation of Htl. The level of RTK activity is crucial for wrapping as wrapping glia lacking *htl* function show no wrapping and wrapping glia with an excess *htl* function show increased wrapping where glial processes invade the fascicle of eight photoreceptor axons [46,51].

High levels of FGF-receptor signaling also activate the expression of the homeodomain protein Cut that is sufficient to induce the formation of glial processes [39,46,51,52]. Thus, FGF-receptor signaling intensity appears to set the start of differentiation. The execution of different wrapping modes is then likely regulated by axon–glia adhesion (Figure 3 and Figure 4). Wrapping glial cells express the Ig domain protein Borderless, that binds its neuronally expressed binding partner Turtle, another Ig domain protein. This interaction is needed for subsequent ensheathment, but the exact mechanism on how this interaction leads to glial wrapping or whether the Borderless–Turtle interaction affects the level of FGF-receptor signaling is presently unknown [53,54]. 

Quite similar to the visual system, FGF-receptor signaling also affects glial differentiation in the larval nerves, where reduction in Htl activity also results in poorly differentiated wrapping glia [23]. Whether wrapping of axons in larval nerves also requires the Borderless–Turtle interaction is not yet known. 

## 8. Larval Wrapping Glial Differentiation Depends on Several RTK Signaling Systems

Morphogenesis of wrapping glia in larval abdominal nerves not only requires FGF-receptor signaling, but also depends on activity of the epidermal growth factor (EGF) receptor [6]. The *Drosophila* EGF receptor can be activated by four different ligands (Spitz, Gurken, Keren, and Vein). Of these ligands, the Neuregulin ortholog Vein plays a role in wrapping glial differentiation in the larva [6]. Vein is generated in glial cells suggesting that an autocrine activation of the EGF receptor is needed to fully control glial differentiation. Interestingly, in mice, Neuregulin signaling also induces myelin formation in Schwann cells, and in injury models, Neuregulin is provided in a cell-autonomous manner to trigger remyelination [55]. 

An additional receptor tyrosine kinase was recently identified in an RNAi-based screen [56]. Knockdown of the Discoidin domain receptor (Ddr) in wrapping glial cells reduced the ensheathment of axons. As Discoidin domain receptors are collagen receptors [57], Ddr may be activated by extracellular matrix proteins such as the type XV/XVIII collagen Multiplexin [56]. The role of extracellular matrix is also emphasized by the severe wrapping glial cell defects caused by integrin knockdown in larval nerves or the optic stalk [58,59]. 

In conclusion, a plethora of extracellular signals act on different RTKs that direct the differentiation of wrapping glia via the Ras signaling cascade. An increase in RTK activity leads to hyperwrapping or increased differentiation of wrapping glial cells characterized by exuberant amounts of glial membranes. In the optic stalk, these glial processes invade photoreceptor axon fascicles [46]. In the larval nerve, they can form multilayered membrane stacks [6,23,46]. Similarly, expression of activated Ras in wrapping glial cells causes the formation of an excess of glial membranes in larval nerves [6]. 

The different RTK signaling cascades might all impinge on a common differentiation path (Figure 5) or might control specific aspects of glial differentiation. Possibly, activation of the FGF receptor, the EGF receptor, and the Discoidin domain receptor establishes a specific temporal RTK activation pattern that is needed to instruct differentiation. This strategy is used during development of the R7 photoreceptor cell, where the activation of the EGF receptor and the activation of the Sevenless RTK is needed for proper differentiation [60]. Alternatively, the activation of different RTK families not only triggers the canonical Ras signaling pathway, but also differentially activates other transduction pathways, for example, via phosphatidylinositol-3-kinase (PI(3)K). How the complex interplay of the different signaling cascades controls glial wrapping differentiation will require further investigation.

## 9. Axon–Glial Interaction to Ensure Correct RTK Signaling Activity

Of the three RTK systems known to act in peripheral wrapping glia, two, Htl and Ddr, are controlled non-autonomously by secreted ligands [46,56]. The EGF receptor, in contrast, appears to be activated in an autocrine manner [6]. Thus, one possible mechanism controlling the extent of wrapping would be that glial cells stop EGF-receptor activity, and thus halt glial growth by stopping the production of the activating ligand vein. Alternatively, adhesion systems may negatively influence RTK signaling activity. Fasciclin 2 (Fas2) and its mammalian orthologue, the neural cell adhesion molecule (NCAM), are both able to suppress EGF-receptor signaling [61,62]. The notion that Fas2 might have signaling functions independent of adhesion is also corroborated by recent genetic analyses [63]. Fas2 is expressed by motor neurons and localizes in a graded manner along their axons. Here, it controls migration of peripheral glial cells that express a different Fas2 isoform [64,65]. In larval stages, Fas2 might block EGF-receptor activity in wrapping glia upon contact with axons. Thus, the single glial wrap of motor axons could be controlled by Fas2-mediated suppression of EGF-receptor activity. 

Other cell-surface proteins known to inhibit EGF-receptor function are Kekkon1 and Echinoid [66,67]. The transmembrane protein Kekkon1 is strongly expressed by many neurons and can directly interact with the EGF receptor [66,68,69]. Similarly, the Ig domain-containing protein encoded by *echinoid* (*ed*) negatively regulates the EGF-receptor signaling during *Drosophila* development [70]. Echinoid functions as a homophilic adhesion molecule. It can also show heterophilic interaction in trans with *Drosophila* Neuroglian, which leads to inhibition of EGF-receptor signaling [67,71,72,73]. Interestingly, Echinoid can also interact with Turtle which affects glial wrapping [53,70,74]. In addition, gap junctions might contribute to increased glia–glia cell adhesion [75,76,77]. Together, these interactions might provide examples on how adhesion systems may locally regulate signaling cascades to control the exact wrapping mode (Figure 3).

An alternative mode to suppress EGF-receptor activity is via the Notch receptor. RTK signaling and Notch show an antagonistic relationship in many developmental contexts [78,79,80,81,82,83,84] (Figure 5). In *Drosophila*, Notch signaling is activated by binding of Delta or Serrate, which triggers the proteolytic cleavage of Notch and the release of the cytoplasmic domain (N^ICD^) that acts as a transcription factor in the nucleus [85,86]. In the embryonic nervous system, Notch affects the generation of glial cells and their migration along axon trajectories [19,87,88,89]. 

**Figure 5 cells-12-02553-f005:**
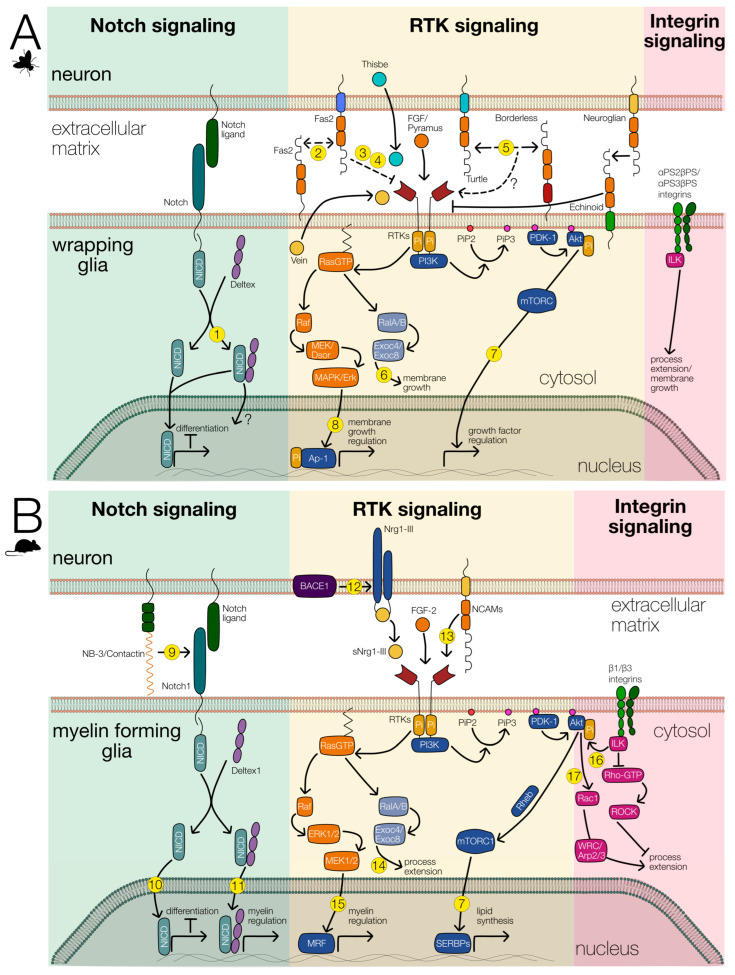
Signaling pathways involved in wrapping glial differentiation. The main signaling systems involved in wrapping glial differentiation are shown. Notch (green background), RTK (yellow background), and integrin signaling (red background). The interaction of these signaling pathways likely regulates glial differentiation in *Drosophila* (**A**) as well as in vertebrates (**B**). References for known interactions are indicated by numbers. (**1**) [90],(**2**) [65], (**3**) [62], (**4**) [63], (**5**) [54,91], (**6**) [92], (**7**) [93], (**8**) [94], (**9**) [95], (**10**) [96], (**11**) [97], (**12**) [98], (**13**) [99], (**14**) [100], (**15**) [101,102], (**16**) [103], (**17**) [104].

During embryonic development, astrocytes and ensheathing glia originate from the same progenitor cell. The differentiation of astrocytes requires FGF-receptor activity [20], which presumably activates the transcription factor Pointed [19]. Together with Notch signaling, this activates Prospero expression, that is needed for astrocyte development [19]. Thus, Notch activity counteracts the development of ensheathing glial cells in the ventral nerve cord of the larva. Interestingly, however, Notch and its interaction partner Deltex, which functions in myelin regulation in mammals, are both strongly expressed in fully differentiated ensheathing glia [105] (Figure 5). The exact role of Notch as well as of Deltex still needs to be elucidated. Possibly, as in the tracheal system, Notch is activated by FGF-receptor signaling via Pointed and subsequent stimulation of Delta expression. Furthermore, Notch might then negatively regulate the expression of FGF-8 ligands, as described for the tracheal system [106]. 

## 10. Comparison with Axon Wrapping Glial Cells in Vertebrates

In vertebrates, axonal wrapping is carried out by oligodendrocytes in the CNS or by Schwann cells in the PNS. Both cell types identify large caliber axons and then form myelin sheets around them. Fascicles of small diameter axons are engulfed by non-myelinating Schwann cells in the so-called Remak fibers [63]. When Schwann cells encounter axons, they first decide whether to form myelin or non-myelinating Remak fibers. This process is called radial sorting and requires signaling proteins like integrins and Ral GTPases [25,107,108,109,110] (Figure 5B). Here, large diameter axons are selected by single Schwann cells that subsequently generate a myelin sheath around one axonal segment. The decision to myelinate is directed by the axon via the presentation of differential amounts of the signaling protein Neuregulin I type III (NRG1), which is a membrane-bound EGF-like protein [111,112]. NRG1 binds to the EGF receptor ErbB3 and induces the formation of an ErbB2/ErbB3 heterodimer that triggers Schwann cell differentiation via Ras signaling [113,114,115]. Myelination in the CNS by oligodendrocytes is not controlled by ErbB signaling, but rather requires FGF-receptor signaling [116]. Similar to ErbB, however, FGF receptor 2 controls myelin thickness via the Ras-MAPK signaling together with protein kinase B (PKB)/Akt signaling [117,118,119,120,121] (Figure 5B). Thus, as found in *Drosophila*, RTK signaling is crucial for wrapping glia differentiation in vertebrates. 

In vertebrates, Notch signaling is also intimately associated with glial development (Figure 5). In the mammalian PNS, Notch signaling regulates the transition from Schwann cell precursors to Schwann cells, controls Schwann cell proliferation, and acts as a brake on myelination, possibly by counteracting RTK activation [122,123]. Similarly, in the CNS, Notch signaling promotes generation of oligodendrocyte precursor cells while inhibiting their further differentiation into myelinating oligodendrocytes [124]. However, Notch also controls some aspects of oligodendrocyte differentiation via its interaction with Contactin-like GPI-anchored proteins [95,125,126], which participates in positioning of voltage-gated ion channels along the axonal membrane [127,128]. The role of Notch and Contactin-like proteins in positioning voltage-gated ion channels has not yet been elucidated in *Drosophila*. Importantly, Notch activity frequently counteracts the activity of the receptor tyrosine kinase signaling cascade [129,130,131]. This further suggests that a close interaction of these two signaling networks is crucial for proper differentiation of myelin forming glial cells [80,132]. 

In conclusion, based on the evolutionary conservation of signaling cascades involved in glial differentiation in flies as well as mammals, we anticipate that *Drosophila* will be instrumental to define further relevant gene functions regulating glial differentiation. In *Drosophila*, recent genetic screens have already revealed several cell-surface proteins that are involved in neuron–glia interaction.

With the ever improving genetic and cell biological tool box, we expect that more of these proteins will be identified, and their role in glial differentiation will be unraveled. Despite the fact that flies do not form compact myelin, they can still form myelin-like structures. The hunt is now on to identify wrapping, and thus myelin promoting genes. These genes could further provide new targets for treating devastating neurodegenerative diseases such as Charcot Marie Tooth disease or multiple sclerosis.

## Figures and Tables

**Figure 1 cells-12-02553-f001:**
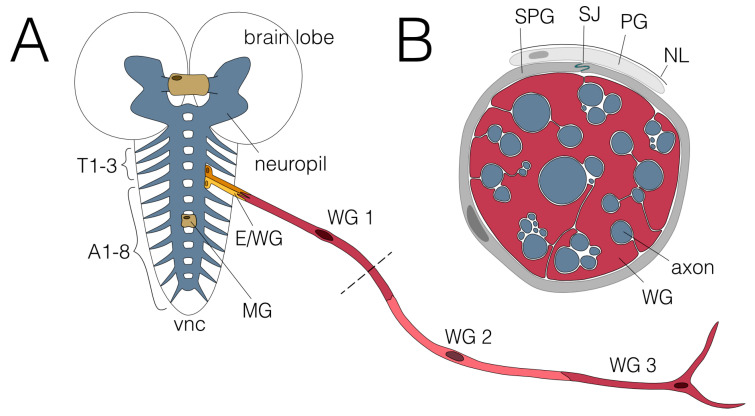
Glial wrapping of axons in the larval nervous system. (**A**) Schematic view of the third instar larval nervous system showing the CNS and one abdominal nerve. The CNS comprises two brain lobes and the ventral nerve cord (vnc). No wrapping of axons has been observed in the neuropil (blue), where in addition to axons, all dendrites and synapses are found. The axons of the two commissures found in each neuromere and the large brain commissure are wrapped by midline glial cells (MG; beige). Axons connect the CNS with the periphery via the three thoracic (T1-T3) and eight abdominal nerves (A1-A8). Two ensheathing/wrapping glial cells enwrap axons connecting the CNS with the PNS in each hemineuromere (E/WG; shades of yellow). Axons within the abdominal nerves are enwrapped by three wrapping glial cells (WG1-3; shades of red). The dashed line indicates the cross section shown in (**B**). (**B**) Cross section of an abdominal nerve. Wrapping glial cells (WG; red) wrap axons (blue) individually or in fascicles. Subperineurial (SPG; dark grey) and perineurial glial cells (PG; light grey) built the blood–brain barrier which is sealed by septate junctions (SJ; green) formed by SPGs. The nervous system is covered by a neural lamella (NL).

**Figure 2 cells-12-02553-f002:**
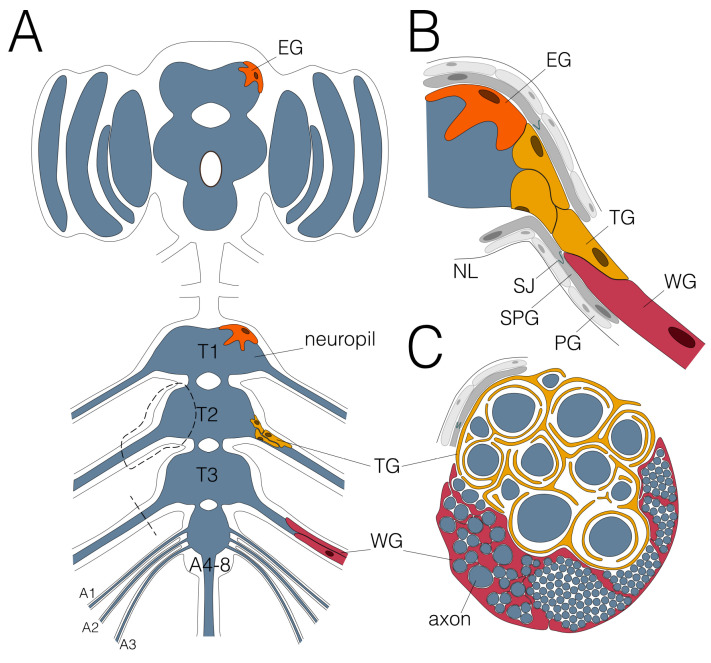
Glial wrapping of axons in the adult *Drosophila* nervous system. (**A**) Schematic view of an adult CNS. Note that the abdominal neuromeres are fused. Dashed areas are shown in (**B**,**C**). Only the leg nerve pairs are indicated, for complete description please see [13]. (**B**) Ensheathing glial cells (EG; orange) cover the neuropil (blue) and wrapping glial cells (WG; red) enwrap axons in peripheral nerves. The tract glial cells (TG; yellow) cover the axons at the PNS–CNS interface. They form extensive processes that establish a lacunar system which collapses more distally to generate myelin-like sheets. (**C**) Schematic cross section of an adult nerve. The adult nervous system is sealed by the blood–brain barrier, comprised by perineurial (PG; light grey) and subperineurial (SPG; dark grey) glial cells which built septate junctions (SJ; green). The nervous system is covered by the neural lamella (NL).

**Figure 3 cells-12-02553-f003:**
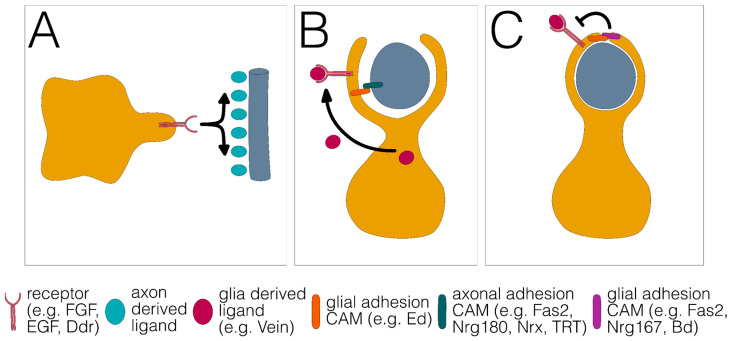
Three distinct phases of glial wrapping. (**A**) Glial growth towards and along the axon. (**B**) Growth around the axon. (**C**) Stop of glial growth around the axon results in a single glial wrap. Different signaling systems and adhesion proteins are listed below.

**Figure 4 cells-12-02553-f004:**
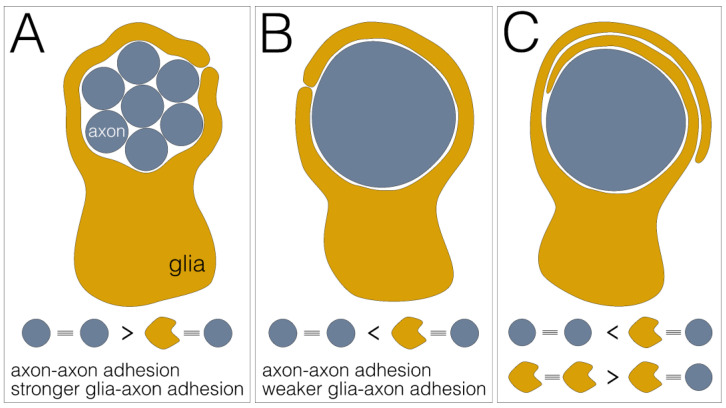
Differential adhesion as glial wrapping mechanism. (**A**) Engulfment of a fascicle comprising multiple axons (blue) by a single glial sheath (yellow). This wrapping mode is established when axon–axon adhesion is stronger than glia–axon adhesion. (**B**) Individual wrapping of a single axon. This wrapping mode occurs when axon–axon adhesion is weaker than glia–axon adhesion. (**C**) Multilayered glial membrane sheets around an axon. This wrapping mode occurs when axon–axon adhesion is weaker than glia–axon adhesion and glia–glia adhesion is stronger than glia–axon adhesion.

## Data Availability

Not applicable.

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
