# Peer review of "Signaling Pathways Controlling Axonal Wrapping in Drosophila"

_cells, 2023, doi:10.3390/cells12212553_

Round 1
Reviewer 1 Report
Comments and Suggestions for Authors
This is a comprehensive review manuscript describing recent advances in our understanding of axonal wrapping by glial cells in Drosophila. The manuscript includes informative sections focusing on anatomical and molecular details, as well as the proposal of new mechanisms of glial regulation and intriguing comparative analysis. Although the manuscript is logically and well-structured, the authors could consider the following suggestions to improve the flow, depth, and the rigorous consideration of additional work related to this field.
In the main text, the authors mentioned “Seven principal glial cell types are known in the larval nervous system”, but they only clearly described the three wrapping types. Although it is clear that this is the focus of the review, the reader would benefit from some additional information of the other glial cell types and their role in the blood brain barrier in the main text, especially since they are depicted in the figures and mentioned in the figure legends. The authors could consider introducing astrocytes here, since this glial cell type suddenly appears in the second half of the manuscript in the sections involving FGF.
In the legend of figure 1, the authors say “Axons enter the CNS via the three thoracic (T1-T3) and eight abdominal nerves (A1-A8).” which is correct for afferent sensory axons, but these nerves also contain efferent motor axons that shouldn’t be ignored.
The property of polyploidy of some glial cells is an interesting one, but it is just superficially mentioned in only one section of the manuscript. It would be beneficial to the reader to clarify whether this is only restricted to a single glial cell type (wrapping glia) and how it contributes to the uniqueness of the described larval and adult glial cells.
It would be helpful to the reader to include a detailed labeling of the nerves of the adult CNS according to the nomenclature established in Court et al., 2020 and also to incorporate this into the main text to provide clarity. In fact, the authors say “the optic and wing nerve both harbor only sensory axons.”, which is confusing, as e.g. the wing nerve is part of the anterior dorsal mesothoracic (ADM) nerve that also contains motor axons (King and Tanouye, 1983). This may help clarify when the authors sometimes describe entire peripheral nerves (“leg nerves”) and sometimes individual nerve branches (“wing nerve”) interchangeably.
Whereas the authors understandably focus on three adult nerves that newly form during pupal stages, (leg, optic, and wing nerves), it would be helpful to mention to what extent the other peripheral nerves show a similar structural organization as the ones described in great detail.
In the first paragraph of the section “Wrapping axons in the central nervous system”, the authors should clarify which of the notable cases of CNS axon wrapping happen only in larvae vs. only adult, as the second paragraph clearly focuses on adult anatomy but this is not clear for the first one.
In the second paragraph of the section “Wrapping axons in the central nervous system”, the authors describe how a high density of voltage-gated sodium channels is present at the axon initial segment of motor axons at the CNS/PNS boundary of adult flies. This discussion would be more complete if previous work is included that characterize the location of the spike initiating zone in a single adult flight motoneuron within distal part of the dendritic tree (Kuehn and Duch, 2013) and how this is contrasted with the location of the tract glia described in this review manuscript and in Rey et al., 2023.
Whereas the authors describe the presence of glial cells at CNS/PNS transition zone, little is discussed about the presence of glial cells at the other end of the nerve (either at the NMJ or at sensory somata). The authors could consider including some description at these locations to provide a more complete picture of the start and end of axonal wrapping.
The authors should consider the suggestion to change the order of figs. 3 and 4, and move the first two paragraphs of the section “Different phases during wrapping glia differentiation” to a later position in the manuscript, since the current fig. 3 deals mostly with molecules which are further elaborated in the following sections and the current fig. 4 is somehow in between but does not necessarily depend on fig. 3.
In the model described in fig. 4C, the authors describe that “when glia-glia adhesion is equal or stronger than axon-glia adhesion, wrapping will not be stopped and multilayered glial membrane sheets will be formed”. The authors could consider discussing the alternative possibility to this scenario that enhanced glial growth could occur independent of axon-glia interactions, forcing the glial membranes in a cell autonomous way to extend on top of each other beyond interactions with the underlying axon.
For fig. 5, the authors could consider including Kuzbanian (the fly homolog of BACE1, which is depicted in 5B) as the secretase producing Notch-ICD in glial cells.
Comments on the Quality of English LanguageVery minor typos detected (e.g. "serval" instead of "several")
Author Response
We are thankful for the time and the excellent comments of the reviewers and have addressed all individual points. Below we specify what we did for every single comment. Our answers are written in BLACK the original review comments in GREY.
This is a comprehensive review manuscript describing recent advances in our understanding of axonal wrapping by glial cells in Drosophila. The manuscript includes informative sections focusing on anatomical and molecular details, as well as the proposal of new mechanisms of glial regulation and intriguing comparative analysis. Although the manuscript is logically and well-structured, the authors could consider the following suggestions to improve the flow, depth, and the rigorous consideration of additional work related to this field.
In the main text, the authors mentioned “Seven principal glial cell types are known in the larval nervous system”, but they only clearly described the three wrapping types. Although it is clear that this is the focus of the review, the reader would benefit from some additional information of the other glial cell types and their role in the blood brain barrier in the main text, especially since they are depicted in the figures and mentioned in the figure legends. The authors could consider introducing astrocytes here, since this glial cell type suddenly appears in the second half of the manuscript in the sections involving FGF.
We have introduced all glial cell types now.
In the legend of figure 1, the authors say “Axons enter the CNS via the three thoracic (T1-T3) and eight abdominal nerves (A1-A8).” which is correct for afferent sensory axons, but these nerves also contain efferent motor axons that shouldn’t be ignored.
This is of course true and we corrected this accordingly.
The property of polyploidy of some glial cells is an interesting one, but it is just superficially mentioned in only one section of the manuscript. It would be beneficial to the reader to clarify whether this is only restricted to a single glial cell type (wrapping glia) and how it contributes to the uniqueness of the described larval and adult glial cells.
We now mention the two other larval glial cell types that become polyploid but have not further commented on this - as we agree - very interesting finding. Since no further report addresses the role of ploidy in larval glial cells (e.g. it might be linked to survival/death of these glial cells) we have not added further speculations.
It would be helpful to the reader to include a detailed labeling of the nerves of the adult CNS according to the nomenclature established in Court et al., 2020 and also to incorporate this into the main text to provide clarity. In fact, the authors say “the optic and wing nerve both harbor only sensory axons.”, which is confusing, as e.g. the wing nerve is part of the anterior dorsal mesothoracic (ADM) nerve that also contains motor axons (King and Tanouye, 1983). This may help clarify when the authors sometimes describe entire peripheral nerves (“leg nerves”) and sometimes individual nerve branches (“wing nerve”) interchangeably.
The reviewer is correct, we oversimplified and added further explanations.
Whereas the authors understandably focus on three adult nerves that newly form during pupal stages, (leg, optic, and wing nerves), it would be helpful to mention to what extent the other peripheral nerves show a similar structural organization as the ones described in great detail.
We think this is beyond the scope of this review, where our main point was to compare adult nerves with motor axons to those nerves that lack a motor component. Moreover, the knowledge on other adult nerves is scarce.
In the first paragraph of the section “Wrapping axons in the central nervous system”, the authors should clarify which of the notable cases of CNS axon wrapping happen only in larvae vs. only adult, as the second paragraph clearly focuses on adult anatomy but this is not clear for the first one.
This is an excellent point and we have added this information.
In the second paragraph of the section “Wrapping axons in the central nervous system”, the authors describe how a high density of voltage-gated sodium channels is present at the axon initial segment of motor axons at the CNS/PNS boundary of adult flies. This discussion would be more complete if previous work is included that characterize the location of the spike initiating zone in a single adult flight motoneuron within distal part of the dendritic tree (Kuehn and Duch, 2013) and how this is contrasted with the location of the tract glia described in this review manuscript and in Rey et al., 2023.
We added this information.
Whereas the authors describe the presence of glial cells at CNS/PNS transition zone, little is discussed about the presence of glial cells at the other end of the nerve (either at the NMJ or at sensory somata). The authors could consider including some description at these locations to provide a more complete picture of the start and end of axonal wrapping.
We have not provided any further information because clear knowledge on glial contribution at adults NMJs is missing. More specifically, it is not always clear whether “wrapping” is an engulfment of subperineurial glia, a blood-brain barrier function, or whether indeed process wrapping occurs.
The authors should consider the suggestion to change the order of figs. 3 and 4, and move the first two paragraphs of the section “Different phases during wrapping glia differentiation” to a later position in the manuscript, since the current fig. 3 deals mostly with molecules which are further elaborated in the following sections and the current fig. 4 is somehow in between but does not necessarily depend on fig. 3.
We had a long discussion about the order of figures and have at the end decided to present the order as is. The reason is that we first mention just recognition of the axon which results in wrapping. The second figure addresses the important issue that wrapping can have distinct qualities. E.g. wrapping of a fascicle, wrapping of single axons or wrapping by multiple layers. This to us is better mentioned second. Thus, we decided to leave the original order.
In the model described in fig. 4C, the authors describe that “when glia-glia adhesion is equal or stronger than axon-glia adhesion, wrapping will not be stopped and multilayered glial membrane sheets will be formed”. The authors could consider discussing the alternative possibility to this scenario that enhanced glial growth could occur independent of axon-glia interactions, forcing the glial membranes in a cell autonomous way to extend on top of each other beyond interactions with the underlying axon.
The reviewer is correct. We added this possibility to the text. Finally it should be noted that glial growth could also occur in a purely cell autonomous way. In case of no axon-glia adhesion, growth is independent of axon-glia interaction and would result in stacked glial cell layers.
For fig. 5, the authors could consider including Kuzbanian (the fly homolog of BACE1, which is depicted in 5B) as the secretase producing Notch-ICD in glial cells.
We have not included this. The role of the BACE1 homolog in Drosophila is unclear. Kuzbanian is a metalloproteinase with different functions and is not the BACE1 ortholog (Bolkan et al 2012). The Drosophila homolog, BACE, is required for APP cleavage and in the absence of BACE expression in neurons, lamina glia die. The phenotype of BACE mutants has not be determined for any wrapping glial cells (Bolkan et al 2012).
Comments on the Quality of English Language
Very minor typos detected (e.g. "serval" instead of "several")
Corrected.
Reviewer 2 Report
Comments and Suggestions for Authors
This review by Baldenius et al is very well written. There is no major concern. The only suggestion is that the term "knockdown" would not be appropriate in the context when the authors are talking about endogenous signaling mechanisms (page 9, line 15). This should be replaced by suppression or inhibition.
Author Response
Comments and Suggestions for Authors
This review by Baldenius et al is very well written. There is no major concern. The only suggestion is that the term "knockdown" would not be appropriate in the context when the authors are talking about endogenous signaling mechanisms (page 9, line 15). This should be replaced by suppression or inhibition.
We are thankful for this suggestion and changed it accordingly.
Reviewer 3 Report
Comments and Suggestions for Authors
In this review, Baldenius et al. first describe the different glial subtypes that wrap axons in Drosophila at larval and adult stage, in the central nervous system as well as in periphery. Then, they propose that the wrapping is done in three successive phases and that the strength of the axon-glia interaction determine the type of wrapping. At last, the authors describe the signalling pathways essential for the wrapping in Drosophila and make a comparison with the pathways activated in mammalian axonal wrapping.
The information provided in the manuscript are informative and give a good overview of the wrapping actors and mechanisms.
The following minor points need to be completed or revised.
The authors need to provide more precision of the definition of wrapping. How does it differ from the interaction between neurons and astrocyte-like glia in the neuropile or from the wrapping of the neuropile by ensheathing glia?
Page 5 part “Different phases during wrapping glia differentiation”: in the two first paragraphs, the authors “propose” three phases for axonal wrapping. Are these just propositions or is there any hard evidences supporting the three phases? Can the authors provide some reference/example?
In the text, the authors mention the possibility that Notch activation negatively regulates FGF signalling (page 11, 1st paragraph after Figure 5’s legend). This link could be added on Figure 5A.
Page 12: the authors mention that Notch counteract the RTK signalling cascade in mammals. Some examples should be provided.

Comments on the Quality of English LanguageDrosophila should systematically be in italic.
The authors should avoid the use of future tense in the manuscript (abstract, page 1 last paragraph, page 6 paragraph 2, legend figure 4…).
Page 1, 2nd paragraph: “several” is misspelled (“serval”).
page 8 paragraph 3: the authors should precise the organism in which Neuregulin induces myelin formation.
Page 8 paragraph 5: this sentence should be rephrased: “Upon RTK activation, exuberant amounts of glial membranes are formed that in the optic stalk invade photoreceptor axon fascicles or in the larval nerve lead to multilayering of glial mem-branes (Franzdóttir et al., 2009; Kottmeier et al., 2020; Matzat et al., 2015)”.
Page 8 paragraph 6: “The discrimination of this possibility and the understanding of the complex interplay of the different signalling cascades controlling glial wrapping will require further investigation.”: I am not sure what possibility are the authors referring to.
Page 8: the following sentence is not necessary: “We need to understand how axons and glial cells communicate to activate or terminate RTK activity.”
Page 11 the acronym of Neuregulin I type III use later in the text should be indicated between parenthesis after the name of the protein.
Author Response
In this review, Baldenius et al. first describe the different glial subtypes that wrap axons in Drosophila at larval and adult stage, in the central nervous system as well as in periphery. Then, they propose that the wrapping is done in three successive phases and that the strength of the axon‐glia interaction determine the type of wrapping. At last, the authors describe the signalling pathways essential for the wrapping in Drosophila and make a comparison with the pathways activated in mammalian axonal wrapping. The information provided in the manuscript are informative and give a good overview of the wrapping
actors and mechanisms.
The following minor points need to be completed or revised.
The authors need to provide more precision of the definition of wrapping. How does it differ from the interaction between neurons and astrocyte‐like glia in the neuropile or from the wrapping of the neuropile by ensheathing glia?
The reviewer is absolutely correct. We added the information stating that we consider wrapping as the planar growth of glial processes around axons or axon fascicle, which eventually results in insulation and the generation of specific compartments.
Page 5 part “Different phases during wrapping glia differentiation”: in the two first paragraphs, the authors “propose” three phases for axonal wrapping. Are these just propositions or is there any hard evidences supporting the three phases? Can the authors provide some reference/example?
This is indeed a proposition of us. There are no references that could yet be added
In the text, the authors mention the possibility that Notch activation negatively regulates FGF signalling (page 11, 1st paragraph after Figure 5’s legend). This link could be added on Figure 5A.
It is tempting to add this interesting link, but as this has been so far demonstrated only for tracheal cells we did not include it in Figure 5A which focusses on signaling cascade regulating wrapping glia development.
Page 12: the authors mention that Notch counteract the RTK signalling cascade in mammals. Some examples should be provided.
We have added three more references.
Formatting points:
Drosophila should systematically be in italic.
Done
The authors should avoid the use of future tense in the manuscript (abstract, page 1 last paragraph, page 6 paragraph 2, legend figure 4…).
Removed as much as possible.
Page 1, 2nd paragraph: “several” is misspelled (“serval”).
Changed.
page 8 paragraph 3: the authors should precise the organism in which Neuregulin induces myelin formation.
Added.
Page 8 paragraph 5: this sentence should be rephrased: “Upon RTK activation, exuberant amounts of glial membranes are formed that in the optic stalk invade photoreceptor axon fascicles or in the larval nerve lead to multilayering of glial mem‐branes (Franzdóttir et al., 2009; Kottmeier et al., 2020; Matzat et al., 2015)”.
Done.
Page 8 paragraph 6: “The discrimination of this possibility and the understanding of the complex interplay of the different signalling cascades controlling glial wrapping will require further investigation.”: I am not sure what possibility are the authors referring to.
We are thankful for this suggestion. The sentence indeed did not make sense and we changed it in the revised version.
Page 8: the following sentence is not necessary: “We need to understand how axons and glial cells communicate to activate or terminate RTK activity.”
Removed.
Page 11 the acronym of Neuregulin I type III use later in the text should be indicated between parenthesis after the name of the protein.
Done.
Round 2
Reviewer 1 Report
Comments and Suggestions for Authors
The authors have satisfactorily addressed the concerns raised during the first round of reviews. Accordingly, this new version of the manuscript is clearer to the reader.